# Construction of Macroeconomic Uncertainty Indices for Financial Market Analysis Using a Supervised Topic Model †

**Kyoto Yono \*, Hiroki Sakaji, Hiroyasu Matsushima, Takashi Shimada and Kiyoshi Izumi**

School of Engineering, The University of Tokyo, Tokyo 113-8654, Japan; sakaji@sys.t.u-tokyo.ac.jp (H.S.); matsushima@sys.t.u-tokyo.ac.jp (H.M.); shimada@sys.t.u-tokyo.ac.jp (T.S.); izumi@sys.t.u-tokyo.ac.jp (K.I.)
\*   Correspondence: d2016kyono@socsim.org
†   This paper is an extended version of our paper published in the 7th International Conference on Smart Computing and Artificial Intelligence, SCAI 2019, Toyama, Japan, 7–12 July 2019.

**Abstract:** The uncertainty in the financial market, whether the US—China trade war will slow down the global economy or not, Federal Reserve Board (FRB) policy to increase the interest rates, or other similar macroeconomic events can have a crucial impact on the purchase or sale of financial assets. In this study, we aim to build a model for measuring the macroeconomic uncertainty based on the news text. Further, we proposed an extended topic model that uses not only news text data but also numeric data as a supervised signal for each news article. Subsequently, we used our proposed model to construct macroeconomic uncertainty indices. All these indices were similar to those observed in the historical macroeconomic events. The correlation was higher between the volatility of the market and uncertainty indices with larger expected supervised signal compared to uncertainty indices with the smaller expected supervised signal. We also applied the impulse response function to analyze the impact of the uncertainty indices on financial markets.

**Keywords:** uncertainty; economic policy; text mining; topic model

## 1. Introduction

Macroeconomic uncertainty is a factor that influences the purchase and sale of financial assets. For investors, the macroeconomic uncertainty is the degree of unpredictability for the future direction of the economy, ranging from several topics, the monetary and fiscal policies in each country and the trade friction between any two countries.

Bank of America Merrill Lynch surveys institutional investors all over the world every month. The survey is about the institutional investor's view of the world economy and its asset class allocation. When the uncertainty about a certain country is high, the institutional investor reduces the weight of the risky assets of a certain country and allocate to the safe assets such as bonds and cash.

For example, when the uncertainty about the EU economy is high and the investors are uncertain about the economic growth in the country except for the US, they allocate their assets to US assets and increase the safe assets such as bonds and cash. On the contrary, when the uncertainty about a certain country disappears, the investors buy the assets of a certain country and reduce the safe assets such as bonds and cash.

### 1.1. Measurement of the Macroeconomic Uncertainty

In a modern economic environment, several macroeconomic uncertainties are observed to co-exist; further, the investors can improve their investment strategies if they can quantitatively classify the

uncertainty based on its source and measure the uncertainty. They can hedge the risk associated with high macroeconomic uncertainty. Additionally, investors may also utilize the uncertainty associated with stress testing.

Historical volatility of a financial asset is partially influenced by the uncertainty (Chuliá et al. 2017). One alternative way to measure the magnitude of uncertainty is to measure the magnitude of historical volatility. However, if assets are influenced by several macroeconomic uncertainties or if volatilities are influenced by demand and supply, measuring only volatility is insufficient to evaluate the uncertainty.

Recently, an alternative method has been proposed to measure the macroeconomic uncertainty using a text mining method. (Baker et al. 2016) developed an approach to measure the policy of economic uncertainty index . (Manela and Moreira 2017) developed the news-implied volatility . Further, we will introduce the related studies in the subsequent section.

### 1.2. Our Contributions

The objective of this study is to construct the macroeconomic uncertainty indices based on the news text. We proposed an extended topic model using both news text data and numeric data as a supervised signal for each news article.

For our research, we used the supervised Latent Dirichlet Allocation (sLDA) and the uncertainties generated without the usage of pre-defined words for each uncertainty (sLDA is one of the topic models discussed in Section 5).

One of the benefits of our model is that our model is able to show the market impact of each uncertainty by using VIX index as a supervised signal for sLDA model. The market impact is not automatically estimated in other models. However, as the results of our model parameter inference, estimated parameter eta is computed as the average magnitude of VIX for each uncertainty index. The relation between estimated parameter eta and market impact is also discussed in Section 6.3. Further, We detail the macroeconomic uncertainty index analysis with historical macroeconomic events. Additionally, we conducted qualitative as well as quantitative analyses of the selected uncertainty indices.

## 2. Related Works

### 2.1. Research Related to Volatility

In financial markets, the correlation between market historical volatility and uncertainty is very high. When uncertainty is high, market historical volatility is also high. (Chuliá et al. 2017) used the stock price to construct the uncertainty index of the market. They separate variation (market volatility) into expected variation and uncertainty (unexpected variation). Many previous studies have performed market volatility prediction using numeric data. (Castelnuovo and Tran 2017) used Google Trends data to forecast weekly volatility of stock markets. (Manela and Moreira 2017) used the news text to predict the market volatility.

### 2.2. Research Using Text Data

(Baker et al. 2016) introduced Economic Policy Uncertainty Index which is constructed by using the news text data. They used pre-defined words for three categories ("policy", "economic", and "uncertainty") to count a number of news articles that contain these words. Moreover, they conducted monthly aggregate seasonal adjustments. This Economic Policy Uncertainty Index had been explored further by many studies. (Jin et al. 2019) analyzed the relation between stock price crash risk and uncertainty index. In addition, (Pástor and Veronesi 2013) explored how stock prices respond to political uncertainty, (Brogaard and Detzel 2015) used uncertainty index to forecast market returns. Not only from the financial market side, but also from the economic side, the uncertainty index have been studied. For example, (Gulen and Ion 2016) analyzed the relationship between the uncertainty

index and corporate investment. (Bachmann et al. 2013; Fernández-Villaverde et al. 2015) analyzed the economic activities using the uncertainty index. Furthermore, (Bloom 2014) discussed the stylized facts about uncertainty index.

Many research have extended Backer's methodology to other countries' news text data and built Economic Policy Uncertainty Index for specific countries. (Arbatli 2017) created the Japan Policy Uncertainty Index, (Manela and Moreira 2017) developed the Belgium Policy Uncertainty Index. (Azqueta-Gavaldon 2017) created Economic Policy Uncertainty in the UK. (Jin et al. 2019) constructed Chinese economic policy uncertainty in similar method as (Baker et al. 2016).

Furthermore, an extension of uncertainty index using different data is also discussed. (Husted et al. 2019; Saltzman and Yung 2018) extracted certain texts from Federal Reserve Beige Books. (Bloom 2014) construct the World Uncertainty Index based on news for 143 individual countries. (Baker et al. 2019) construct Equity Market Volatility tracker. (Hamid 2015) developed uncertainty index by using Google Trend data.

*2.3. Topic Model*

LDA is one of the most popular topic models proposed by (Blei et al. 2003) on the basis that all the documents are a mixture of latent topics and that each topic is a probability distribution with respect to words. Text analysis based on the bag of words representation has a high dimension (the number of dimensions is the number of distinct words in the corpus). By using topic models the number of dimensions of text can be reduced the dimensions equals the number of topics. Many researchers applied the LDA model to Financial News. (Hisano et al. 2013) used LDA to classify Business News. (Shirota et al. 2014) used LDA to extract financial policy topics from the Policy Board of the Bank of Japan financial policy meeting proceedings. (Mueller and Rauh 2018) use extended LDA model to classify Federal Open Market Committee text and discuss the FOMC Communications on US Treasury Rates. (Kanungsukkasem and Leelanupab 2019) introduced an extended LDA model called FinLDA. (Thorsrud 2018) use LDA to classify newspaper topic and construct a daily business cycle index. (Larsen and Thorsrud 2019) use LDA to classify newspaper topic and predict economic variables.

*2.4. Topic Model Applying to Uncertainty Index*

(Azqueta-Gavaldón 2017) used topic models to build an uncertainty index. He used topic models to separate news text into 30 topics and selected topics which is equivalent to categories identify by (Baker et al. 2016). He successfully replicates Backer's Economic Policy Uncertainty Index in a less costly and more flexible way.

(Rauh 2019) used regional newspapers to build uncertainty at the regional level. He uses the topic model to separate news text into 30 topics and extract 5 topics (Independence, Energy, Investment, Federal, Government, Topic index) as an uncertainty index.

The difference between this study and the previous studies is that we constructed multiple uncertainty indices using a supervised LDA model. The description of the supervised LDA is provided in Section 5. By using the supervised LDA model, uncertainty indices that have a strong relationship with market volatility can be separated from others.

## 3. Datasets

In this study, we extract the uncertainty index based on the news text and apply the renowned topic model to the news text for classifying the topic of uncertainty. News text is considered to be the dataset for the model. Besides, numeric data is also considered to be the dataset as the supervised signal for each article.

*3.1. Text Data*

The text data were obtained from the Japanese Reuters news articles, and we extracted the global economy news article from the Reuters website. In total, we collected 33,000 articles from August

2009 to November 2019. More than ten global economy news articles are published on the Reuters website per day, and each article contains an average of 1200 words. The news articles on the global economy category are observed to focus on the economic events and monetary policy on each major country. Furthermore, comments and columns of economists are available on the website. We used the news articles from the global economy category because the text corpus should contain articles related to global economic uncertainty and not individual firm issues or market price movement. This condition is almost the same as Baker's conditions of extracting the article which contains terms related to "economic" and terms related to the "policy" category.

*3.2. Numeric Data*

We selected the volatility index (VIX) as the supervised signal for the sLDA model. The VIX concept is used to formulate theoretical expectations of the volatility implied by the S&P 500 index, which is disseminated and calculated on a real-time basis by the Chicago Board Options Exchange. VIX is an index that allows investors to measure uncertainty for future market trends. Thus, we collect the VIX dataset daily similar to the interval during which we collect the text data.

## 4. Materials and Methods

Figure 1 divides the entire process into three parts. The first part is the prepossessing of the input data, and the second part is the topic classification that is performed using sLDA. The final part is to measure uncertainty index. We use topic distribution for each document and normalize in each month.

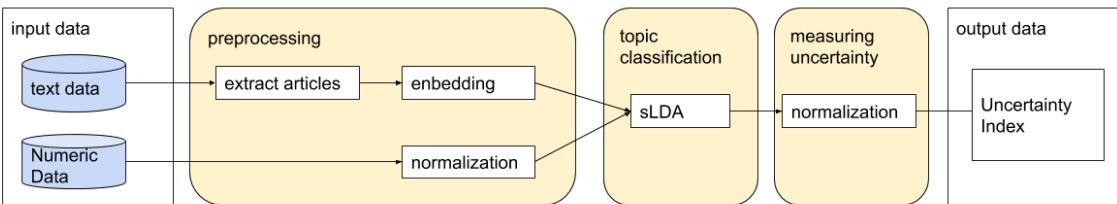

**Figure 1.** Overall process.

*4.1. Text Data Preprocessing*

After obtaining Reuters news articles from the global economy category of the Reuters'website, we extracted articles that explained the uncertainty. First we define uncertainty terms (Table 1). For all articles, we extracted articles that contain any of the uncertainty terms. Because some Reuters articles contain articles that are not related to uncertainty but views about the economic forecast, we should eliminate these articles from the corpus. This condition is the same as Baker's conditions of extracting the article which contains terms related to the "uncertainty" category.

**Table 1.** Term sets for uncertainty.

| Uncertainty Terms |
| --- |
| "Uncertainty" or "Uncertain" |
| "Indetermination" or "Indeterminate" |

Subsequently, we conduct Japanese language morphological analysis and extract nouns using Mecab (Japanese Part-of-Speech and Morphological Analyzer). As a result of text data preprocessing, the corpus contains 3115 documents and approximately 2.95 million words with 1786 distinct terms.

### 4.2. Numeric Data Prepossessing

For numeric data, we use the VIX index as the supervised signal for sLDA. We conducted normalization to convert the VIX index to exhibit a zero average and a standard deviation of one. Further, we use the converted VIX index as the supervised signal for each article.

### 4.3. Topic Classification

The topic classification is conducted using sLDA. The supervised signal for each article is the converted VIX having the same date the article was published.

### 4.4. Uncertainty Measurement

After the topic classification was completed using sLDA, $\theta_{d=1..M,k=1...K}$ the probability of topic k occurring in document d is inferred. Here $M$ is the total number of documents and $K$ is the total number of topics. Next, we calculate the average probability topic k occurring in documents for the month $t$ by the following equation.

$$S_{t=1...T,k=1...K} = \frac{1}{n_t} \sum_{d_i \in D_t} \theta_{d_i,k=1...K}. \tag{1}$$

Here, $n_t$ denotes the number of documents at month $t$ and $D_t$ denotes the collection of documents at month $t$. Finally, the score for the uncertainty index of topic k at the month t ($UI_{t,k}$) is the normalized value of $S_{t,k}$. We normalized $S_{t,k=1...K}$ so that the average score of each topic to 100 by the following equation.

$$UI_{t=1...T,k=1...K} = \frac{S_{t=1...T,k=1...K}}{\frac{1}{T}\sum_{t}^{T} S_{t,k=1...K}} \times 100. \tag{2}$$

## 5. Topic Model

*Supervised Latent Dirichlet Allocation*

Ever since LDA was introduced by (Blei et al. 2003), there were many extension models of LDA.

sLDA is an expansion of LDA proposed by (Mcauliffe and Blei 2008). sLDA is a model developed by adding a response variable associated with each document to the LDA model, which jointly model the documents and responses, to find latent topics that will optimally predict the response variables in case of future unlabeled documents.

Figure 2 presents the graphical model representation of sLDA and Table 2 presents the sLDA notations. In our research, the converted VIX index is used as a signal for the article.

**Table 2.** Notations in sLDA.

| Notation | Definition |
| --- | --- |
| $\alpha, \beta$ | Hyperparameters |
| $\varphi$ | The distribution over words |
| K | The number of topics |
| $\theta$ | The document specific topic distribution |
| Z | A topic |
| w | A word in the document |
| N | The number of words |
| M | The number of documents |
| Y | Response variable |
| $\eta, \sigma$ | Hyperparameters for the response variable |

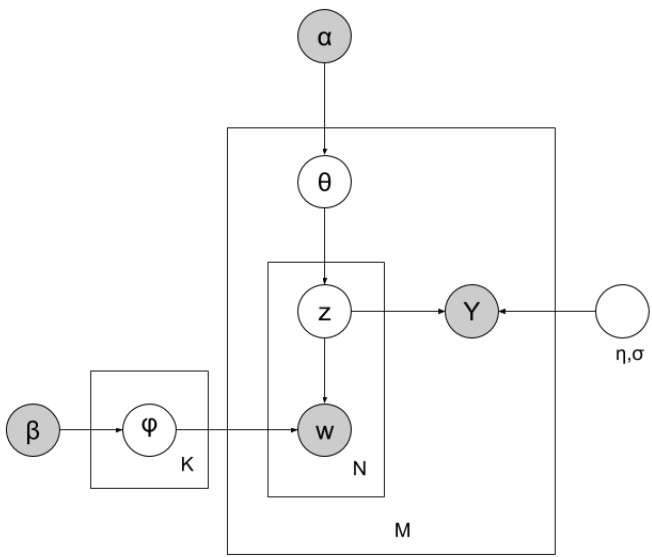

**Figure 2.** The graphical model representation of sLDA.

The generation process in sLDA is consists of following steps:

- For each document $d$, topic distribution $\theta_d$ determined by the following equation, where $\alpha$ is the hyperparameter of Dirichlet distribution.

$$\theta_d \sim Dirichlet(\alpha). \tag{3}$$

- For each topic $k$, word distribution $\varphi_k$ determined by the following equation, where $\beta$ is the hyperparameter of Dirichlet distribution.

$$\varphi_k \sim Dirichlet(\beta). \tag{4}$$

- For each word $w_{d,i}$ in document $d$

  - topic $z_{d,i}$ is sample from distribution by the following equation.

  $$z_{d,i} \sim Multinomial(\theta_d). \tag{5}$$

  - word $w_{d,i}$ is a sample from distribution by the following equation.

  $$w_{d,i} \sim Multinomial(\varphi_{z_{d,i}}). \tag{6}$$

- For each document $d$, response variable $Y_d$ is sample from distribution by the following equation, where $\bar{z}_d := (1/N_d) \sum_{n=1}^{N_d} z_n$.

$$Y_d \sim N(\eta^T \bar{z}_d, \sigma^2). \tag{7}$$

## 6. Results

### 6.1. Topic Classification

We performed topic classification by sLDA using the following parameters: $\alpha = 0.35, \beta = 0.10, \sigma = 1.0, K = 10$. We set the variance of supervised signal $\sigma = 1.0$. This is because we use the normalized VIX index as the supervised signal. As for $\alpha, \beta$ we did parameter search among

$\alpha \in [0.2, 0.25, 0.3, 0.35, 1.0, 3.0, 6.25], \beta \in [0.1, 0.05, 0.01, 0.005]$. We select $\alpha = 0.35, \beta = 0.10$ because the result of topic classification results is clearly separated.

The perplexity of sLDA was calculated by different number of topics ($K$). Figure 3 show the relationship between the perplexity of the model and number of topics.

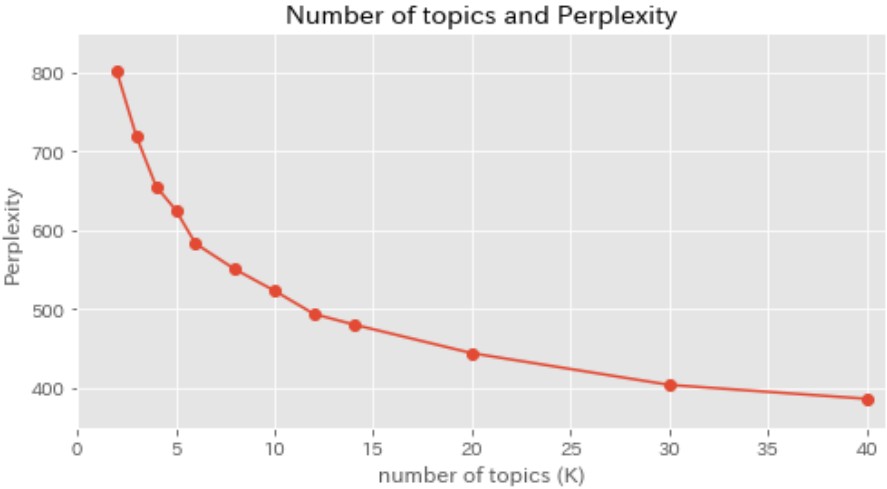

**Figure 3.** Number of topics and Perplexity.

The reason we set $K = 10$ is that even if the number of topics $K > 10$, the perplexity does not change much further but too many numbers of topics are hard to interpret.

The topic classification results obtained using the sLDA model were presented in Table 3.

**Table 3.** Top 10 words for each topic word distribution $\varphi_k$.

| TOPIC 0 | TOPIC 1 | TOPIC 2 | TOPIC 3 | TOPIC 4 |
|---|---|---|---|---|
| Central bank | EU | Action | policy | risk |
| Inflation | Brexit | Thomson | Trump | China |
| President | UK | we | finance | World economy |
| Anticipation | European Union | Norm | government | Point out |
| interest rate | Trade | Principle of trust | administration | Europe |
| Point out | USA | Influence | reform | growth |
| growth | risk | recovery | Parliament | Emerging countries |
| policy | Influence | production | Prime Minister | Influence |
| Inflation | Point out | Economy | Politics | Deceleration |
| Financial policy | investment | Supply | Tax increase | USA |

| TOPIC 5 | TOPIC 6 | TOPIC 7 | TOPIC 8 | TOPIC 9 |
|---|---|---|---|---|
| Greece | BOJ | Dollar | Company | FRB |
| Europe | prices | market | GDP | Rate hike |
| Bank | Relaxation | Euro | economist | FOMC |
| Euro zone | add to | Rise | quarter | Chairman |
| market | Committee | Decline | investment | Shrink |
| support | President | Market price | Anticipation | President |
| Euro | Influence | USA | export | Federal Reserve Board |
| debt | policy | Stock price | Point out | Relaxation |
| Finance | monetary easing | Strategist | Elongation | policy |
| Point out | necessary | Investor | View | market |

For the top words of each topic word distribution $\varphi_k$, we interpret topics as follows. Topic 0 is a topic related to uncertainty on monetary policy in the EU; topic 1 is a topic related to uncertainty on international economic events affecting the world economy; topic 2 is a topic related to the Great East Japan Earthquake. Although some of the most frequent words in topic 2 are related to the disclaimer

in the news, some of the most frequent words are related Great East Japan Earthquake. The time series of topic 2 (Figure 4) shows a peak after the Great East Japan Earthquake in 2011;

Topic 3 is a topic related to uncertainty on fiscal policy in the US; topic 4 is a topic related to uncertainty on economic growth in China and emerging countries; topic 5 is a topic related to uncertainty on financial system risk in the EU; topic 6 is a topic related to uncertainty on monetary policy in Japan; topic 7 is a topic related to uncertainty on financial markets; topic 8 is a topic related to uncertainty on monetary policy economic growth in Japan; topic 9 is a topic related to the uncertainty on monetary policy in the US;

The time series of the monthly average percentage of document topic distribution of each topic is shown in Figure 4.

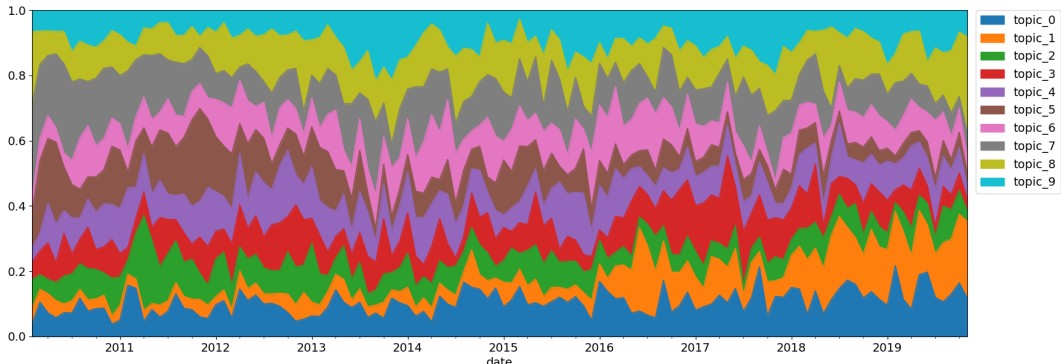

**Figure 4.** Time series of percentage of document topic distribution.

The weight of topic 1 which is a topic related to uncertainty on international economic events affecting the world economy is increasing over time. The weight of topic 5, which is a topic related to uncertainty on financial system risk in the EU, decreases over time. Except for theses topics, almost all topics remain stable. The lowest weight over the entire period is topic 2 which is topically related to the Great East Japan Earthquake.

### 6.2. Uncertainty Indices with Macroeconomic Event

In this section, we consider time-series generated uncertainty indices $UI_{t,k}$ with the related macroeconomic event.

Figure 5 shows the time-series of topic 1 which is related to uncertainty on international economic events affecting the world economy. A: September 2014. The Scottish independence referendum in the UK. B: June 2016. The United Kingdom European Union membership referendum. C: March 2018. US-China trade friction.

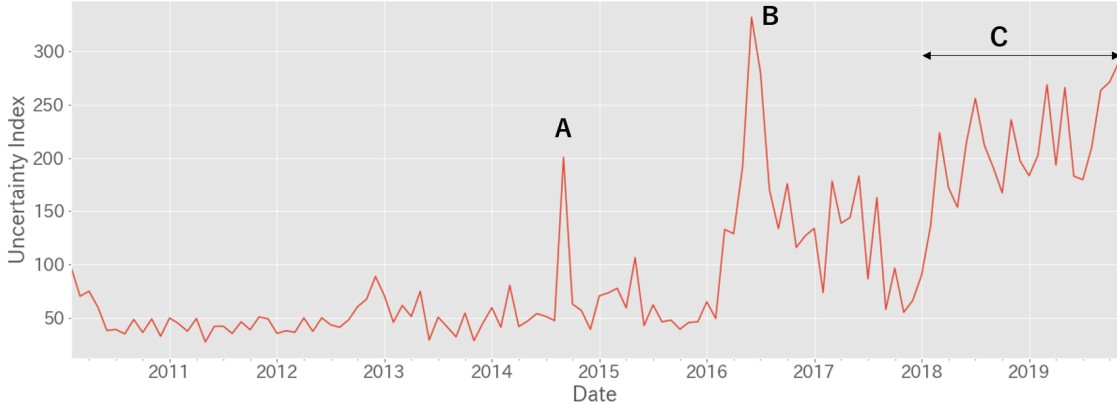

**Figure 5.** Tpoic 1: uncertainty on international economic events affecting the world economy.

Figure 6 shows the time-series of TOPIC 3 which is related to uncertainty on fiscal policy in the US. D: November 2012. Obama re-elected as president. E: November 2016. Trump elected as president.

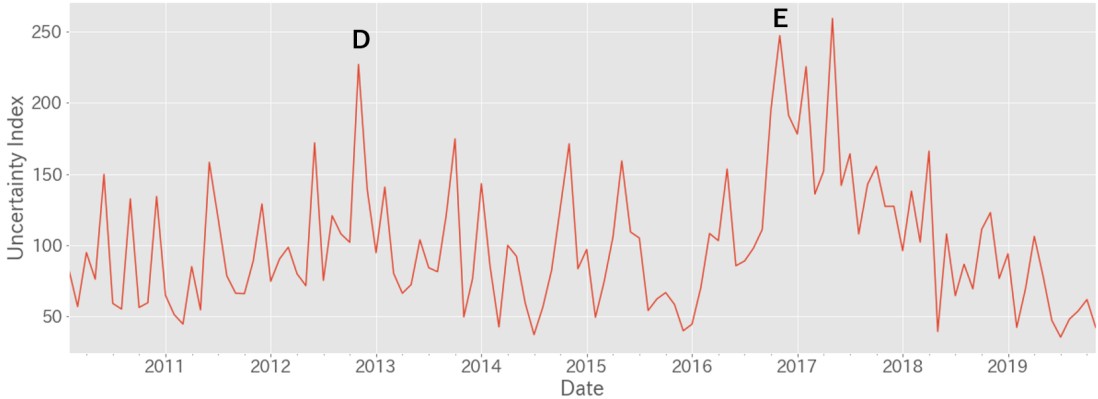

**Figure 6.** Topic 3: uncertainty on fiscal policy in the US.

Figure 7 shows the time-series of TOPIC 4 which is related to uncertainty on economic growth in China and emerging countries. F: January 2011. Economic overheating and concerns about inflation in emerging countries and China. G: August 2012. The slowdown in China's industrial production index. H: March 2014. Ukrainian crisis. I: August 2015. China shock. J: March 2018. US-China trade friction.

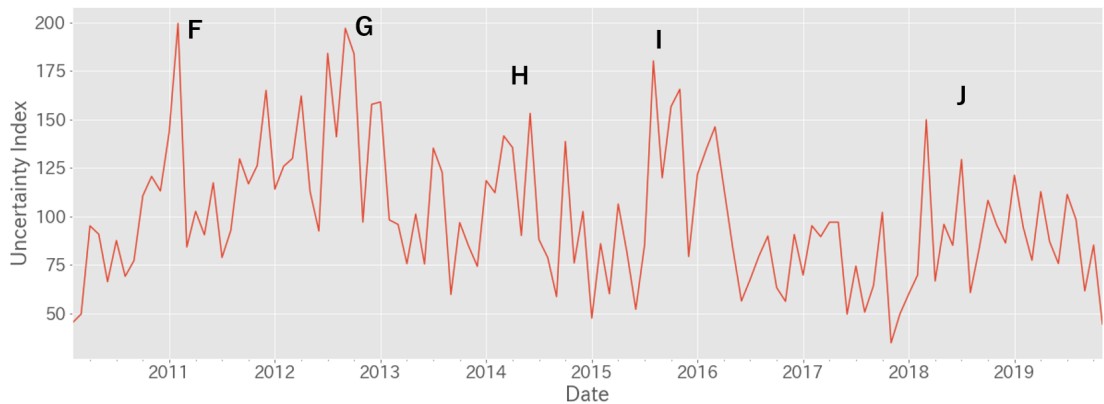

**Figure 7.** Topic 4: uncertainty on economic growth in China and emerging countries.

Figure 8 shows the time-series of topic 5 which is related to uncertainty financial system risk in the EU.

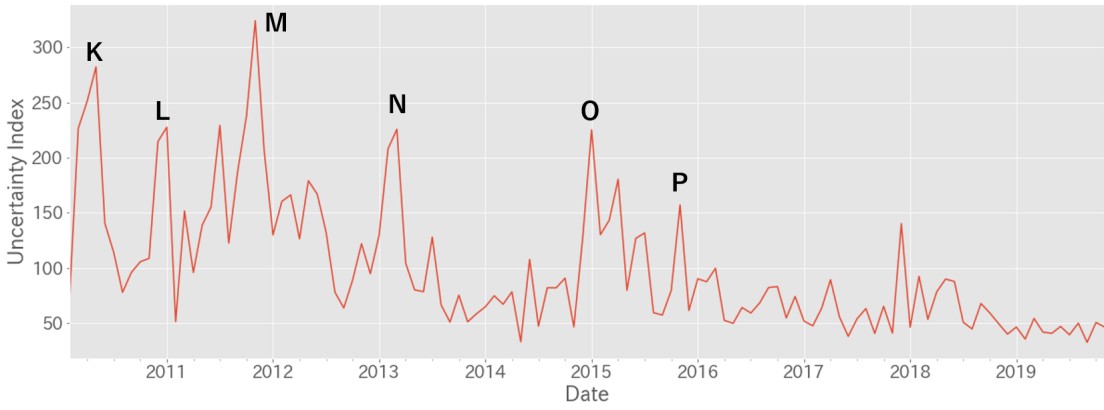

**Figure 8.** Topic 5: uncertainty on financial system risk in the EU.

K: April 2010. Greece debt crisis. L: November 2010. Rating firms downgraded Greek government bonds. M: November 2011. The referendum on accepting support measures from the European Union. N: March 2013. Cyprus shock. O: January 2015. Greek General Election and Uncertainty about the future of negotiations with the EU increased. P: November 2015 Portuguese Government Bonds Excluded from ECB Bond Purchase Program due to Portugal's political situation uncertainty increased.

Figure 9 shows the time-series of topic 6 which is related to uncertainty monetary policy in Japan. Q: April 2012. Bank of Japan increased funds for asset purchases by about 10 trillion yen. R: October 2012. Bank of Japan increased funds for asset purchases by about 10 trillion yen. S: April 2013. Bank of Japan decided to introduce quantitative and qualitative monetary easing policy. T: January 2016. Bank of Japan decided to introduce quantitative and qualitative monetary easing policies with negative interest rates. U: July 2016. Bank of Japan decided additional monetary easing by increasing the ETF purchase amount, etc.

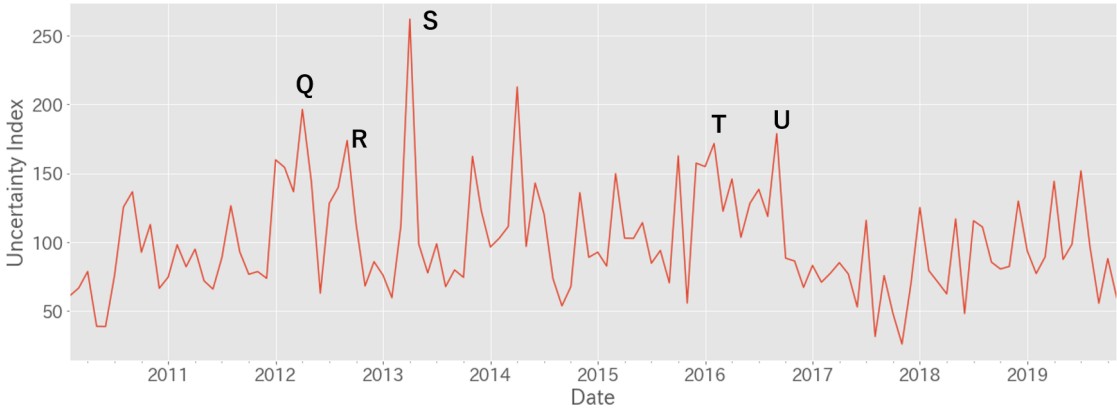

**Figure 9.** Topic 6: uncertainty on monetary policy in Japan.

The country-specific uncertainty index build by Baker's model contains the macroeconomic uncertainties happened in the other countries. For example, economic policy uncertainty (EPU) for Japan contains the macroeconomic uncertainty of European Debt Crisis in 2011 (Arbatli 2017). These macroeconomic uncertainties should not be contained in EPU for Japan but should be only contained in EPU for the EU. As for our model, we use sLDA model to separate the uncertainties based on its topics. the macroeconomic uncertainty caused by the European Debt Crisis only contributes to the TOPIC 5 (uncertainty on financial system risk in the EU), not to the other country-specific uncertainties.

### 6.3. Comparison with Baker's Model

In this section, we present a comparison of the Uncertainty Indices constructed by the proposed model and Uncertainty Index constructed by Baker's model. Due to limited access to the news text corpus, we used the same news text corpus as we used in our model (Section 3.1) to build Baker's model. Against (Arbatli 2017), they used several Japanese newspapers in their research, we used Japanese Reuters news on the Reuters website. We constructed the Japan Economic Policy Uncertainty Index by following the same model in (Baker et al. 2016). The model counts the number of articles that contains three categories Japanese words (economy, policy, uncertainty) as Baker's model. The outputted Japan Economic Policy Uncertainty Index is shown in Figure 10. We compare this Japan Economic Policy Uncertainty Index constructed by Baker's model and Uncertainty Indices constructed by our proposed model.

First of all, we examined the term frequency of the sentences of each peak of the Japan Economic Policy Uncertainty Index by Baker's model (Figure 10 from A to E). Top 10 words with a frequency of occurrence are shown in Table 4 .

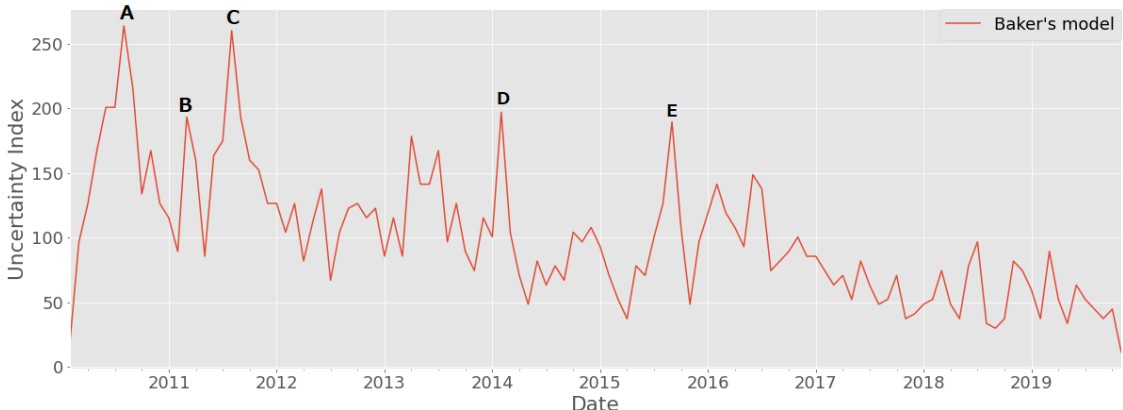

**Figure 10.** Japan Economic Policy Uncertainty Index constructed by Baker's model.

**Table 4.** Top 10 words for each peaks of the Japan Economic Policy Uncertainty Index by Baker's model.

| A | B | C | D | E |
|---|---|---|---|---|
| Future View | Market | Market | Economy | Rate hike |
| Yen Appreciation | Japan | Europe | Emerging countries | Financial market |
| Business condition | Influences | Financial | Market | Stock Price |
| Dollar | Bank of Japan | Yen Appreciation | View | Slow down |
| Market | Rising | Future | Pointing out | Market |
| Economy | Possibility | Business | USA | World economy |
| Downside | Crude oil price | Economy | Business | Future |
| Enterprise | President | World economy | Movement | China economy |
| Countermeasure | FRB | Financial | Mitigation | China |
| Europe | Middle East | USA | Japanese economy | Economy |

As you can see from Table 4, the keywords related to the countries other than Japan ("USA", "Europe", "FRB", "Emerging countries" and "China" etc;) , and keywords related to the market ("Financial market", "Yen Appreciation" and "Dollar" etc;) appeared in the Table.

The major events on each peak are related to not only economic event happened in Japan but also in other countries as well. A: Fiscal and financial instability in Greece and other EU countries B: Great East Japan Earthquake C: The U.S. debt ceiling issue D: The contraction of QE3 in the U.S. and concerns about emerging economies E: Slowdown in China and other emerging economies and resource prices falling

We compared the above results with Table 3 which is the uncertainty index constructed by our proposed model. As for topic 6 which is an uncertainty index related to monetary policy in Japan, the top 10 keywords only contain keywords related to Japan and not keywords related to the countries other than Japan. Similarly, topic 9 which is an uncertainty index related to monetary policy in the US, the top 10 keywords only contain keywords related to the US and not keywords related to the countries other than the US.

As a quantitative comparison between the Japan Economic Policy Uncertainty Index by Baker's model and Japan Economic Policy Uncertainty Index by our proposed model, we calculate the ratio of keywords related to the country, keywords related to the market, keywords related to the other country.

As you can see from Table 5, our model have a high percentage of keywords compared to baker's model. In this section, we compared the Japanese Uncertainty Indices constructed by the proposed model and the Japanese Uncertainty Index constructed by Baker's model. As a result, the index by Baker's model is more likely to include keywords related to other countries than those related to the local country. It can be said that the index by Baker's model includes more uncertainty caused by the global economic factor and market factor than the country factor. Conversely, the index by our

model is divided by topics and the uncertainty index by specific country does not contain elements of uncertainty caused by foreign countries and markets.

**Table 5.** The ratio of keywords related to the country, other countries, and financial market.

|  | Keywords Related to the Country | Keywords Related to Other Countries | Keywords Related to Financial Market |
|---|---|---|---|
| our model | 75.56% | 12.22% | 12.22% |
| baker model | 14.22% | 40.85% | 44.93% |

The effect to impulse response function caused by the difference between index by Baker's model and index by our proposed model is further discussed in Section 6.5.

*6.4. Correlation with Other Indices*

In this subsection, we discuss the relationship with financial market indices, and the relationship with the uncertainty index created by (Baker et al. 2016).

In Table 6, we present $\bar{\eta}$ of each index at first two columns and Pearson correlation coefficient between the macroeconomic uncertainty indices and volatility of financial market indices (US 10-year bond, S&P500, USD/JPY), and VIX index in the last four columns. The volatility of each financial market indices is computed by a standard deviation of the daily return of the target month. Note that $\bar{\eta}$ is estimated eta after inference and indicates the expected value of a supervised signal for each uncertainty indices, which is then normalized VIX index.

**Table 6.** The $\bar{\eta}$ and Pearson correlation coefficient between uncertainty indices and the volatilities of financial market indices.

| TOPIC | | Pearson Correlation Coefficient | | | |
|---|---|---|---|---|---|
| k | $\bar{\eta}$ | USG10 | S&P500 | USD/JPY | VIX |
| 0 | 0.044 | −0.328 | −0.118 | −0.383 | −0.033 |
| 1 | −0.876 | −0.442 | −0.133 | −0.195 | −0.297 |
| 2 | 0.700 | 0.374 | 0.197 | 0.053 | 0.235 |
| 3 | −0.354 | −0.183 | −0.283 | 0.134 | −0.269 |
| 4 | 0.436 | 0.261 | 0.253 | −0.102 | 0.183 |
| 5 | 1.286 | 0.565 | 0.366 | 0.177 | 0.417 |
| 6 | −0.048 | −0.056 | 0.043 | 0.065 | −0.007 |
| 7 | 0.667 | 0.346 | 0.108 | 0.322 | 0.213 |
| 8 | −0.177 | −0.216 | −0.203 | −0.221 | −0.110 |
| 9 | −0.194 | −0.277 | −0.235 | 0.046 | −0.267 |

The results denote that uncertainty indices with larger parameter $\bar{\eta}$ have a higher positive correlation with market volatility and VIX index and uncertainty indices with smaller parameter $\bar{\eta}$ have a higher negative correlation with market volatility and VIX index.

For example, topic 5 which is related to the uncertainty on financial system risk in the EU (Table 3, Figure 8) and topic 7 which is related to the uncertainty on financial markets (Table 3) have larger $\bar{\eta}$ value than the other topics. These topics and volatilities of other financial market indices show a stronger positive correlation.

To the contrary, topic 1 which is related to the uncertainty on international economic events affecting the world economy (Table 3, Figure 5) and topic 3 which is related to the uncertainty on fiscal policy in the US (Table 3, Figure 6) have smaller $\bar{\eta}$ value than the other topics. These topics and volatilities of other financial market indices show a stronger negative correlation.

In addition, topic 0 which is related to the uncertainty on monetary policy in the EU (Table 3) and topic 6 which is related to the uncertainty on monetary policy in Japan (Table 3, Figure 9) has $\bar{\eta}$ value close to 0. These topics and volatilities of financial market indices shows no correlation.

Note that topic 2 which is the topic related to the Great East Japan Earthquake have a higher positive correlation with USG10 volatility than other topics. This is because topic 2 which contains both the disclaimer and uncertainty related to the Great East Japan Earthquake according to topic word distribution and the heightened uncertainty in the Japanese economy caused by the earthquake in 2011 coincided with the fall in U.S. interest rates triggered by the European debt crisis. If we exclude the peak in 2011, the correlation between topic 2 and USG10 volatility decreases from 0.37 to 0.14 and this is lower than other topics.

The relationship between $\bar{\eta}$ and the Correlation coefficient with financial market indices is more clearly shown through scatter plots (Figure 11).

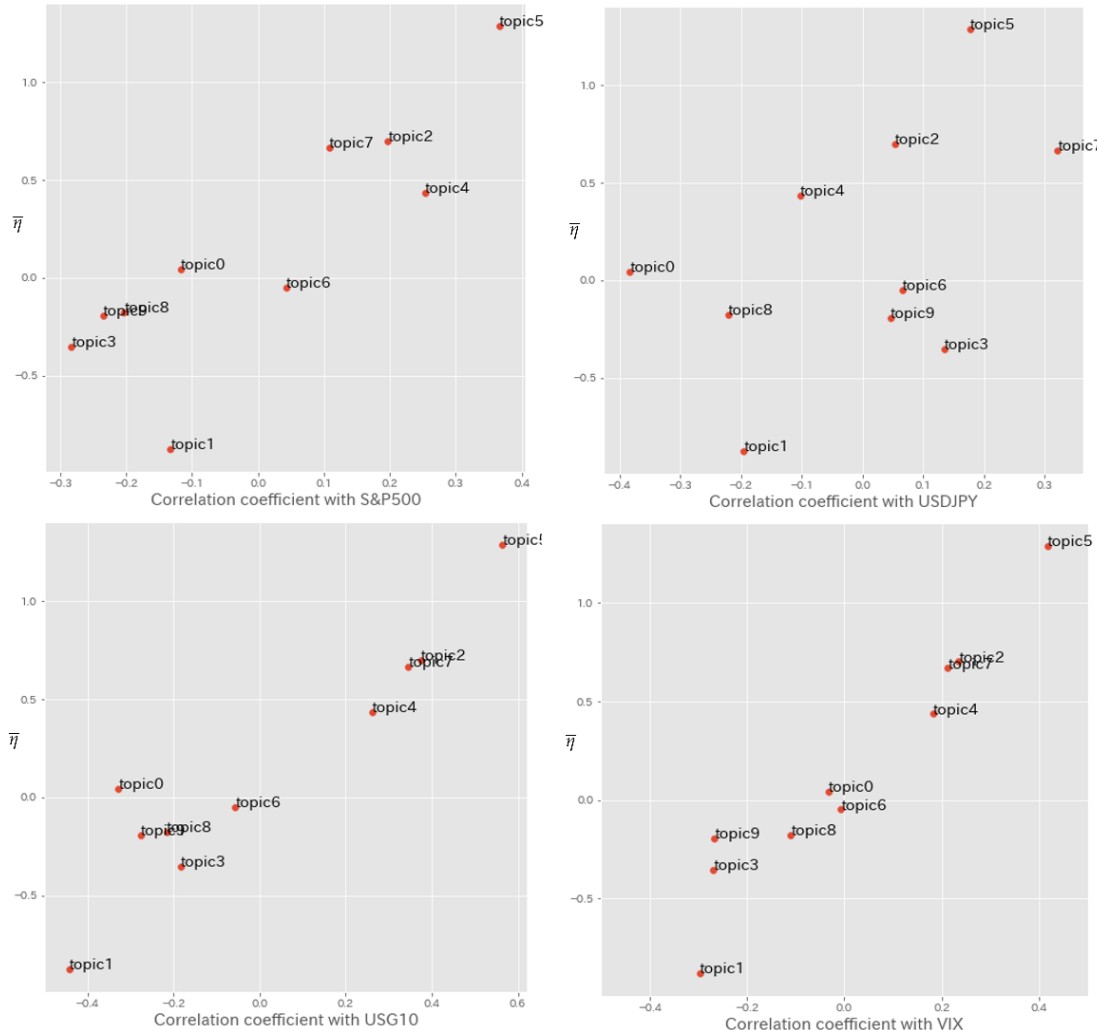

**Figure 11.** The $\bar{\eta}$ and Pearson correlation coefficient between uncertainty indices and market indices.

The results of Table 6 also show that topic 9 which is related to the monetary policy of the US has a negative correlation with the S&P500 index. The reason is although S&P500 is the stock index of the US market, it is influenced by not only the monetary policy of the US but also from other global economic events such as china shock (topic 4), EU debt crisis (topic). This is why topic 9 which is related to the monetary policy of the US has a negative correlation with the volatility of S&P500 index.

In Table 7, we present $\bar{\eta}$ of each indices at the first two columns and Pearson correlation coefficient between the our macroeconomic uncertainty index and four Economic Policy Uncertainty Index (EPUI) by (Baker et al. 2016) in the last four columns.

To the contrary the relationship between $\bar{\eta}$ and Correlation coefficient with market indices (Table 6), the results denote that uncertainty indices with smaller parameter $\bar{\eta}$ have a higher positive correlation

with EPUI and uncertainty indices with larger parameter $\bar{\eta}$ have a higher negative correlation with market volatility and VIX index.

**Table 7.** The $\bar{\eta}$ and the Pearson correlation coefficient between uncertainty indices and Economic Policy Uncertainty Index (EPUI) by (Baker et al. 2016).

| TOPIC | | Pearson Correlation Coefficient | | | |
|---|---|---|---|---|---|
| k | $\bar{\eta}$ | Global | China | UK | US |
| 0 | 0.044 | 0.251 | 0.372 | 0.096 | −0.028 |
| 1 | −0.876 | 0.687 | 0.752 | 0.648 | 0.260 |
| 2 | 0.700 | −0.226 | −0.266 | −0.312 | −0.005 |
| 3 | −0.354 | 0.098 | 0.029 | 0.325 | 0.131 |
| 4 | 0.436 | −0.117 | −0.127 | −0.263 | 0.040 |
| 5 | 1.286 | −0.285 | −0.381 | −0.300 | −0.058 |
| 6 | −0.048 | −0.086 | −0.078 | −0.105 | −0.084 |
| 7 | 0.667 | −0.481 | −0.537 | −0.315 | −0.182 |
| 8 | −0.177 | 0.032 | 0.120 | −0.038 | −0.117 |
| 9 | −0.194 | −0.007 | 0.014 | 0.038 | −0.034 |

Following Figure 12 show the scatter plots between $\bar{\eta}$ and Correlation coefficient with EPUI by (Baker et al. 2016).

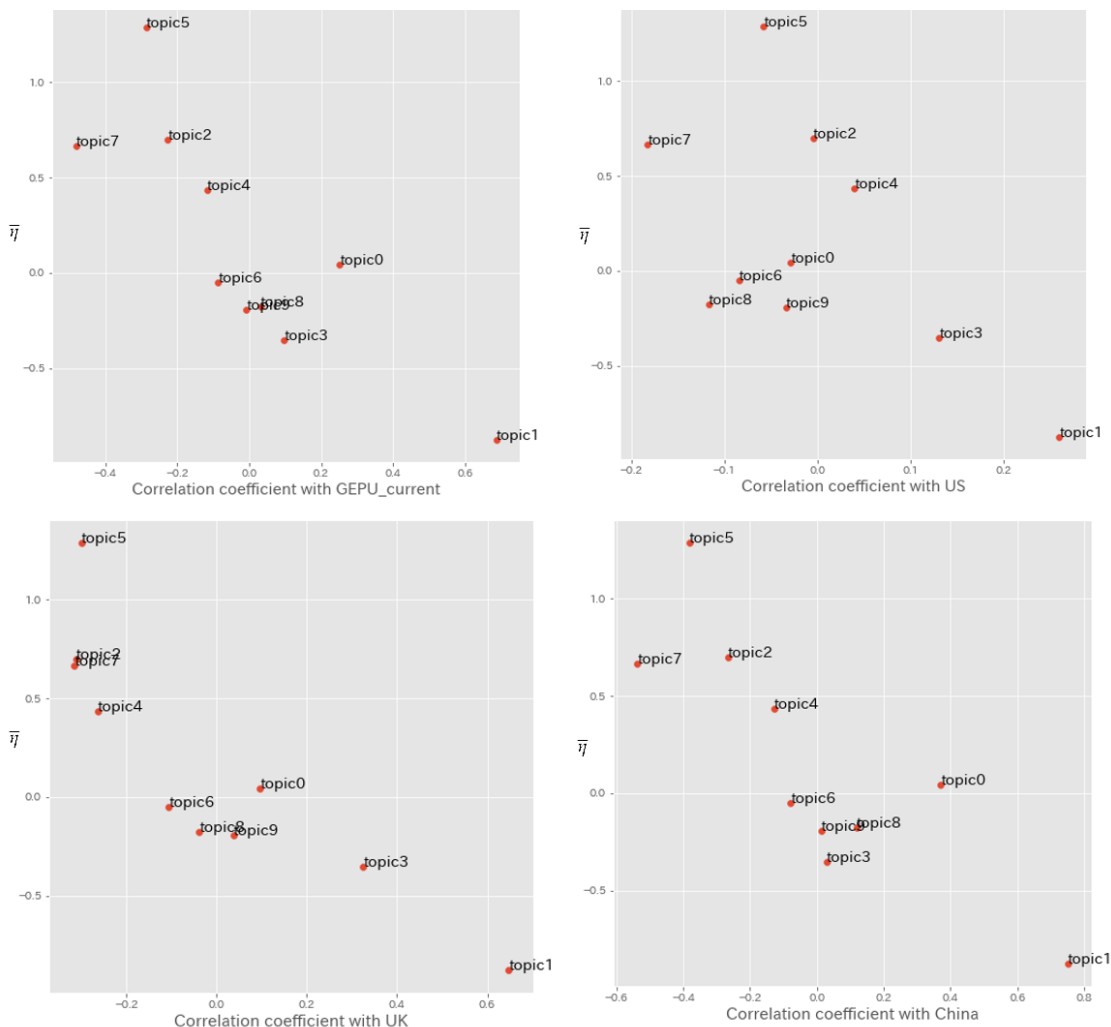

**Figure 12.** The $\bar{\eta}$ and correlation coefficient with EPUI by (Baker et al. 2016).

The results of Table 7 also show that TOPIC 9 which is related to the monetary policy of the US has a very weak correlation with Baker's indices. Within Backer's Economic Policy Uncertainty indices, the percentage of Monetary policy of all Policy Category is less than 30%. This is why the correlation between Backer's Economic Policy Uncertainty and our TOPIC 9 which is related to the monetary policy of the US is very weak.

The above analysis shows:

- by applying sLDA model which uses normalized VIX index as a supervised signal, the model can extract topics highly linked to market volatility (topics 2, 5 and 7).
- extracted topics that are not closely related to the market volatility are highly correlated with the existing uncertainty index (TOPIC 1, TOPIC 3). The market impact of these topics is limited because the market is already been factored into the market.

*6.5. Impulse Response Analysis*

We conducted the var analysis between uncertainty index and macroeconomic by using the Japan Industrial production index (Figure 13).

To analyze the impact of the uncertainty index by our model on the industrial production index, we constructed three VAR models as follows;

- bivariate VAR with one variable being VIX index and the other variable being Japanese Industrial Production
- bivariate VAR with one variable being Japanese uncertainty index based on Baker's model which we build in Section 6.3 and the other variable being Japanese Industrial Production
- bivariate VAR with one variable being the topic-specific indices by our we proposed model and the other variable being Japanese Industrial Production

Due to the limitations of the data, the data period is from 2013/1 to 2019/11. The estimation method is OLS and the identification condition used to compute the impulse response functions is Cholesky decomposition and the variables are in order of uncertainty index and related market variable.

The VIX index has an impact of about 0.4 standard deviations on the industrial production index after four months.

The uncertainty index created based on Baker's model does not show a significant impact. Out of our topic-specific indices, topic 4 (global economy uncertainty index) and topic 6 (Japan uncertainty index) shows a significant impact on Industrial Production Index, with a negative impact comparable to that of the VIX index.

The Japanese uncertainty index constructed by Baker's model had an impact on pushing down the Japanese industrial production index, but it did not show any significance. This is because the uncertainty index by Baker's model fails to break down into country-specific uncertainty as described in Section 6.3.

Although (Arbatli 2017) showed the significance of the Japanese uncertainty index constructed by the Baker's model affecting industrial production index, but we can not show the significance in our analysis with different text corpus and different data period for var model.

With our proposed model, each uncertainty index of a specific country or specific topic is constructed with less news test data compares to (Arbatli 2017), some uncertainty indices show significant impact on the Industrial Production Index.

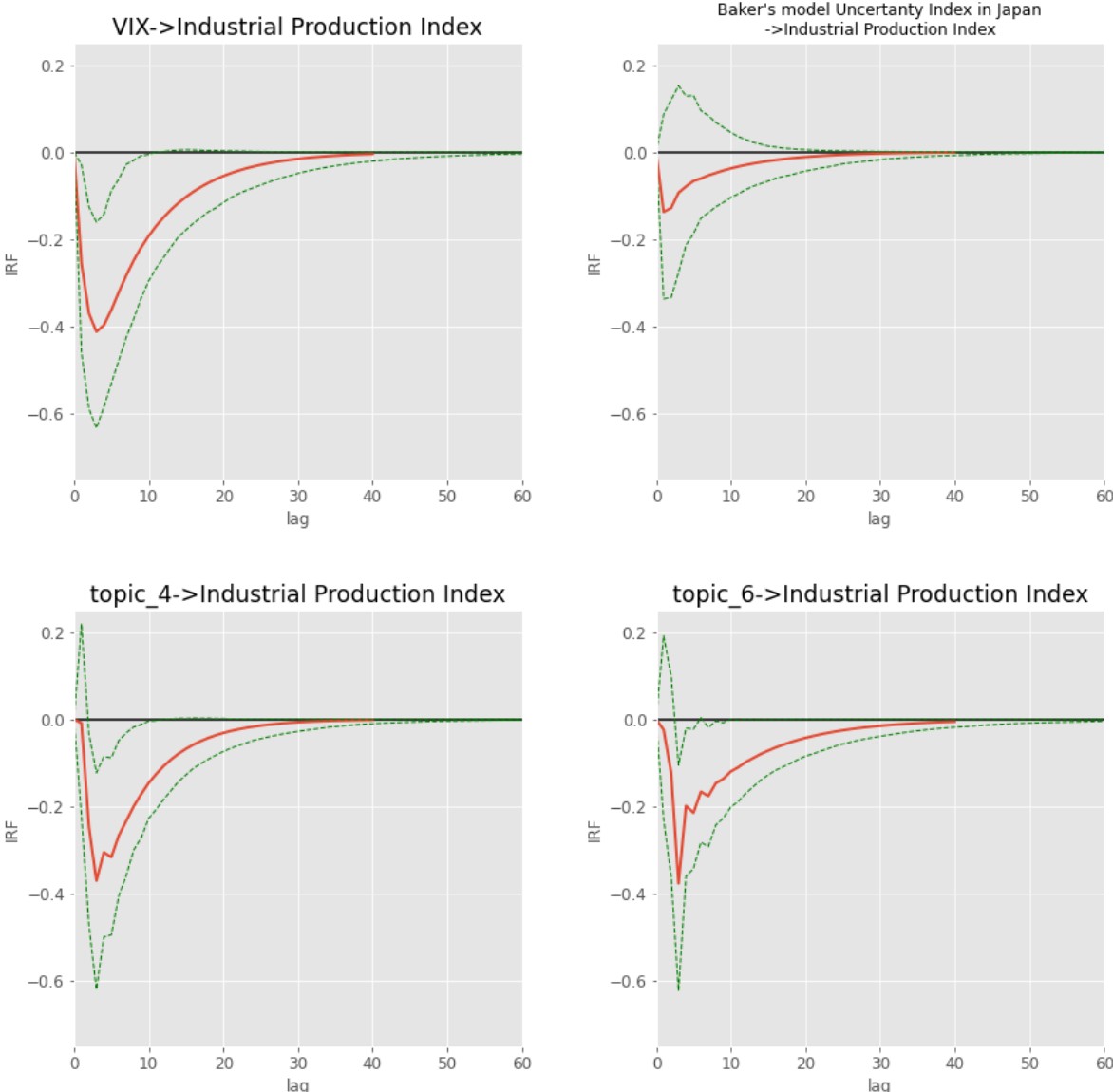

**Figure 13.** Impulse Responses to Unit Standard Deviation Uncertainty Index Innovation.

## 7. Discussion and Conclusions

In this study, we apply the sLDA model to extract uncertainty indices from the news text and the VIX index as a supervised signal. We constructed uncertainty indices based on the topics generated by sLDA. Further, We conducted correlation analysis based on the volatility of the market indices and impulse response analysis based on the related market indices. The results denote that the macroeconomic uncertainty indices with larger parameter $\bar{\eta}$ have a higher positive correlation with financial market volatility and VIX index, which enable sLDA model to extract topics highly linked to market fluctuations.

Currently, our research is conducted by Japanese news articles and it is limited to Reuters News only. In future work, we will expand our news corpus to several sources and also we will conduct the same analysis in the English version of news articles.

**Author Contributions:** Conceptualization, K.Y., H.S., T.S., H.M. and K.I.; methodology, K.Y.; software, K.Y.; validation, K.Y., H.S., T.S., H.M. and K.I.; formal analysis, K.Y.; investigation, K.Y.; resources, K.Y.; data curation, K.Y.; writing—original draft preparation, K.Y.; writing—review and editing, K.Y.; visualization, K.Y.; supervision, K.I. All authors have read and agreed to the published version of the manuscript.

**Funding:** This research received no external funding.

**Conflicts of Interest:** The authors declare no conflict of interest.

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
