# Peer review of "Construction of Macroeconomic Uncertainty Indices for Financial Market Analysis Using a Supervised Topic Model"

_jrfm, doi:10.3390/jrfm13040079_

Round 1

Reviewer 1 Report

The paper is potentially interesting and can be improved to merit a publication. The authors should attempt to address the following points.

The authors need to elaborate more on papers by Azqueta-Gavaldón and Rauh who have used topic models before to develop uncertainty indices. Is there a reason for setting K=10? Have the authors checked by setting K < 10? It would be interesting to see how the uncertainty indices perform by setting K < 10. I am a little concerned about some chosen words in some of the topics: for example, Topic 0: it refers to monetary policy in EU and how "President" is relevant in that? Topic 5: How "Point Out" is important? Similarly, Topic 8 (economist), and so on. Can you explain why Topic 9 which is related to the monetary policy of the US has a negative correlation with the S&P index? Should not the correlation be stronger? Also, the correlation with Baker's indices is very weak. That needs to be better explained.  

Reviewer 2 Report

See referee report.

Reviewer 3 Report

See the attached PDF file.

Reviewer 4 Report

Report on  “Construction of macroeconomic uncertainty indices for financial market analysis using supervised topic model”   This paper proposes a method to construct uncertainty indices using text data. It uses the supervised Latent Dirichlet Allocation method to analysis text data and uses an existing volatility index as the supervised signal. The method generates several topic distributions of texts and the authors show that each topic is related to different types of macroeconomic events.   I think that this is an interesting contribution and has strong potential to be published in JFRM. However the paper contains many unclear points and awkward expressions. I list those points below. The writing must be improved for this paper to be published. I do not think that any additional analysis is necessary; just polishing the paper is required for publication.   Line 22-27 I do not konw whether the claim can be substantiated. References should be given. Or this paragraph can be eliminated.   Line 34 “The easiest way” I wonder it is really the easiest.    Line 56 ``Volatility is a concept similar to uncertainty” This is a finance paper and volatility should be defined precisely. Also “volatility" and “uncertainty” are different things.   Section 5.1 Does the notation for sLDA follow Mcauliffe and Blei (2008). We cannot understand what it means even if we are told that $\alpha$ is a hyperparameter. The notation should be defined or provides a reference in which we can get the definition. Alternatively, because most of the notation in Table 2 do not appear later in the paper, they may not need to be introduced.   Section 6.3 What is the definition of $\bar \eta$?    Line 204  How are the volatilities of USG10, S\&P500 and USD/JPY computed?   Section 6.4 Which kind of VAR is computed? Is it a bivariate VAR(1) with one variable being an uncertainty index and the other variable being a related market variable? What is the estimation method? Is it OLS? What is the identification condition used to compute the impulse response functions? Is it the Cholesky decomposition? If so, how are the variables ordered?

Round 2

Reviewer 2 Report

see report attached.

Reviewer 3 Report

The revised paper is much improved relative to the earlier version of the paper. One my comments, which is also raised by Reviewer 2, that has not been explicitly addressed in the revised version of the article is the comparison of the method proposed in the present paper with the existing method(s) in the literature. In my first round of review, I noted that:

" The bulk of the paper focuses on the discussion and implementation of the proposed uncertainty index. However, it is not clear what the main advantage(s) of the proposed uncertainty index is relative to existing uncertainty indices in the literature. Specifically, relative to Baker et al. (2016, Quarterly Journal of Economics) and some others discussed in Sections 2.2 and 2.3. To address this concern, the authors need to provide comparisons of performance of the proposed uncertainty index with those of the (couple) existing indices using simulated and/or real data. This would help to establish whether computational complexity of the proposed method justifies its benefits (if any). "

In the revised version of the paper, the authors provided some discussions defending their method. But, I think adding explicit comparison of the proposed and existing method(s) would elevate the contribution of the present paper, providing tangible results.

Round 3

Reviewer 2 Report

see report attached.

Reviewer 3 Report

The authors have reasonably addressed my comments, for which I thank them. 

Author Response

Thank you very much for your review. It is very helpful for us to improve the paper.